# Bioactive Components of Pomegranate Oil and Their Influence on Mycotoxin Secretion

**DOI:** 10.3390/toxins12120748

**Published:** 2020-11-27

**Authors:** Ahmed Noah Badr, Hatem Salama Ali, Adel Gabr Abdel-Razek, Mohamed Gamal Shehata, Najla A. Albaridi

**Affiliations:** 1National Research Centre, Department of Food Toxicology and Contaminants, Cairo 12622, Egypt; 2Department of Food Science and Nutrition, College of Food Science and Agriculture, King Saud University, Riyadh 11451, Saudi Arabia; 3National Research Centre, Department of Food Technology, Cairo 12622, Egypt; 4National Research Centre, Department of Fats and Oils, Cairo 12622, Egypt; adelgabr2@gmail.com; 5Department of Food Technology, Arid Lands Cultivation Research Institute, the City of Scientific Research and Technological Application (SRTA-City), New Borg El-Arab 21934, Alexandria, Egypt; gamalsng@gmail.com; 6Department of Physical Sport Science, Nutrition and Food Science, Princess Nourah Bint Abdulrahman University, P.O. Box 84428, Riyadh 11671, Saudi Arabia; naalbaridi@pnu.edu.sa

**Keywords:** pomegranate oil, encapsulation, mycotoxin reduction, bioactive phytochemicals, zearalenone

## Abstract

Pomegranate, similar to other fruits, has juice-extraction by-products. Pomegranate seed oil (PGO) is a non-traditional oil with health benefits, rich in bioactive components. This study was aimed to assess PGO phytochemicals and their influence as bioactive components to reduce mycotoxin secretion. The encapsulation was applied in micro and nanoforms to protect the quality and enhance the efficacy of the oil. The PGO was extracted using ultrasound-assisted methods. Carotenoids, tocochromanols, sterols, phenolic, flavonoid, antioxidant, and antimicrobial activity were determined. The fatty acid profile was analyzed by the GC-MS, while mycotoxin was determined utilizing the HPLC apparatus. The toxicity and protective action of oil were examined using the hepatocytes’ cell line. The resultant oil acts as oleoresin that is rich in bioactive molecules. Phenolics and antioxidant potency recorded higher values compared to traditional vegetable oils, whereas polyunsaturated fatty acids were 87.51%. The major fatty acid was conjugated punicic acid (81.29%), which has high biological effects. Application of the PGO on fungal media reduced aflatoxins secretion up to 63%, and zearalenone up to 78.5%. These results confirm the bio-functionality of oil to regulate the fungal secondary metabolites process. The PGO is a unique prospective non-traditional oil and has several functionalities in food, which achieve nutritional, antioxidant, and anti-mycotoxigenic activities.

## 1. Introduction

Pomegranate fruits (POMFs) are edible plants with a commercial concern, cultivated in the Middle East, Southern-East Asia, and Mediterranean areas. The POMFs are eaten as food due to the premium health impact and nutritional value and are appended in non-food industries to produce dyes, cosmetics, and pharmaceuticals [1]. Besides, pomegranate could be applied in the manufacturing of the concentrates, coloring, and flavoring agents. The pomegranate fruit divided from the inside into juicy clusters uncover groups of gem-like seeds; seeds as byproduct affluent in a special kind of oil could be applied in different manufacturing and food processing.

Recently, the POMFs have acquired renewed global interest, where it possesses more beneficial antioxidant components and is deemed a principal source of the bioactive compounds. In addition, it has been characterized by its richness in functional and nutraceutical substances [2]. POMFs emerged in folk medicine and were recorded with efficacy in preventing diseases. The POMF by-products that resulted from the manufacturing were accredited for antioxidant and antimicrobial activities exhibited by their content of bioactive compounds [3].

Phenolic components are from the category of natural compounds, which has a hydroxyl functional group (-OH) connected to an aromatic hydrocarbon ring. They were classified as natural lipophilic radical scavengers generated to be one of the forms of aegides of the defense system against stressful situations [4]. In addition, flavonoid compounds provide a natural defense against free radicals and pro-oxidants by acting as reductant and radical scavenging [5,6]. The flavonoids have low molecular weight with the prevalent benzopyrone structure. This has illustrated their enormous biological and pharmacological functions. These characteristics represent a better solution to improve food safety due to the impact on microbial contamination.

The major hazard resulted from contamination issues were toxigenic fungi and their related mycotoxins. Contamination is a significant risk that turns food into useless materials and is deemed the critical hazards in several production stages of food, both in pre and post-harvest [7,8]. Because of their growth in food material, it affects the acceptability, toxicity, and sensory of food products. Food contamination may occur due to its infection by harmful toxigenic fungi or their secondary metabolites [4,5]. Aflatoxin B_1_ (AFB_1_), which are excreted by toxigenic fungi and reported with a hepatotoxic impact, is the most hazardous member of the AF family [6]. Natural bioactive components, such as those that existed in the PGO, have achieved biological influences against these types of food hazards. Commonly, phenolic antioxidants necessitate delaying food spoilage, where it possessed antimicrobial activities [4]. In addition, it could reflect an extension of shelf life by minimizing the log CFU of total microorganism count.

Although the POMFs are rich in bioactive components, both the aqueous and oily extracts. Various investigations focused on utilizing aqueous extracts in food applications. However, more studies are requested to investigate the pomegranate oleoresin impact against microbial contamination. The POMF seed contains Punicic acid as a major fatty acid (18:3 omega 5 CLnA) of their oil, which is stated to have many biological effects [8]. In this regard, the present investigation target to predict and illustrate the effect of bioactive components of the PGO against microbial contamination hazard, particularly on limiting mycotoxin production. Besides, the determination of cytotoxicity impacts of the PGO to the cell line, and its in vitro corrective action against the AFB_1_ toxicity using the cell line. Furthermore, to protect the PGO bioactivity, an encapsulation technique has been applied.

## 2. Results and Discussion

### 2.1. Total Phenolic and Flavonoid Content

In this study, total phenolic content (TPC) of PGO was 39.06 ± 1.47mg GAE/g oil, while the total flavonoid content (TFC) was 12.4 ± 0.96 mg CE/g oil (Table 1). The obtained values of the TPC were lower than that were reported for the pomegranate juice [7,8]. The difference in phenolic contents between juice and oil according to their hydrophilicity and attraction to polar solutions.

The focal aspect regarding that; the plant phenolics attracted considerable attention due to their beneficial, functional, and nutritional effects, including antioxidant and antimicrobial activity. Besides, they act as a scavenger of the free radical and prevent cell-damaging [9].

### 2.2. Tocochromanols Composition (Tocols)

Concerning the lipophilic antioxidants (Table 1), the PGO showed a featured amount of total tocochromanols (911.7 µg/g), carotenoids (205.31 µg/g), and sterols (1639.36 mg/100 g). The PGO deems as oleoresin; its contents of tocopherols and tocotrienols derivatives were stunning. The δ-tocopherol was a dominant form (414.6 ± 2.52µg/g), while other types were α-tocopherol (16 ± 2.0 µg/g), β-tocopherol (0.4 ± 0.42 µg/g), and γ-tocopherol (13.7 ± 1.53 µg/g).

Concerning tocotrienols, four types were observed; where γ-tocotrienol (462.7 ± 3.05 µg/g) was more abundant than δ-tocotrienol (2.03 ± 0.21 µg/g), α-tocotrienol (1.6 ± 0.15 µg/g), and β-tocotrienol (0.67 ± 0.12 µg/g). This result is consistent with Beveridge et al. [10], who described that γ-tocotrienol is the main fraction extracted from the pomegranate seeds. Tocochromanols (Vitamin E) are important antioxidants that compose the backbone of the pomegranate oleoresins (crude PGO), the oxidative potency of this extract is mostly based on the presence of such compounds [11].

### 2.3. Phytosterols

The predominant sterol found in the PGO was β-sitosterol, followed by campesterol, while delta-5 avenasterol comes later (Table 1). In contrast, delta-7 stigmasterol, stigmasterol, brassicasterol, and campestanol were detected in low concentrations. This result is consistent with Habibnia et al. [12], who investigate the Iranian PGO varieties. Moreover, Pande and Akoh [13] confirmed β-sitosterol as the principal sterol extracted from pomegranate seeds with concentrations variation. Phytosterols are acting biological functions similar to mammalian cholesterol. It was reported by much less absorbance due to tiny structure differences than cholesterols [14]. It also inhibits the cholesterol absorption through the intestinal that categorized phytosterol as lipid-lowering agents [14], resulting in preventing certain types of cancer as well [15].

### 2.4. Carotenoids

The carotenoids in the PGO have existed in the form of three derivatives (Table 1). The predominant fraction was Lutein (188.3 ± 1.79 µg/g), followed by Zeaxhantin (16.47 ± 0.21 µg/g), then β-Carotene (0.547 ± 0.078 µg/g) came later. On the contrary, the present results differed from those reported by Yoshime et al. [16], who did not detect any carotenoids in the PGO. These differences mentioned connect to the extraction method or fruit variety. According to Chirinos et al. [17], numerous health benefits have been attributed to carotenoids, which are considered a health promotion because of the responsibility for critical biological functions.

### 2.5. Antioxidant Activity

As described in Figure 1, the results showed a radical scavenging activity of the PGO, which was determined as 32.8, 42.2, 50.3, 56.7, and 66% with different concentrations 62.5, 125, 250, 500, and 1000 µg/mL PGO, respectively, while the reducing power activity was determined within the applied concentrations (7.8, 15.6, 31.2, 62.5, 125, 250, 500, 1000 µg/mL). A significant difference (*p* < 0.05) among reducing activities observed at different concentrations (0–1000 µg/mL PGO). The highest reducing power was recorded at 1000 µg/mL of oil concentration (Figure 1). The antioxidant potency illustrates how polyphenols enhanced the efficacy against the free radicals and carcinogenic materials. The reducing capacity of the PGO recorded increasing by the concentration increases (0–1000 µg/mL); this relation is supported by a previous study [18]. It is worth mentioning that there are no published data on PGO reducing power efficacy.

### 2.6. Phenolic Acid Composition

The compounds that present in the PGO are characterized by utilizing the well-defined peaks with maximum absorbance at 284 nm for phenolic acids. The inspection of chromatographic separations enabled us to identify 20 compounds (Table 2). The quantitative determination of each compound was recorded. Gallic acid was the main component found in the PGO (784.13 ± 0.25 mg/100g), followed by Catechin (396.63 ± 0.45 mg/100g sample). To the best of our knowledge, this is the first time in which twenty phenolic compounds analytically were identified in the PGO oleoresin.

Different ranges of phenolic fractions, including chlorogenic, rutin, quercetin, protocatechuic, kaempferol, lutein, 4-hydroxybenzoic, vanillic, and vanillin were detected, while sinapic acid was not detected in the PGO. Nearly, most of these compounds investigated for their biological and pharmacological properties and literature data had described a range of interesting activities: antibacterial, antiviral, antioxidant, anticancer, anti-inflammatory, anti-aging, and hepato-protective. In recent decades, an increasing number of publications on phenolic compound bioactivity support these data. This finding demonstrates the importance given to understanding the biological efficacy behind these compounds, for instance, regarding their antioxidant, anticancer, antitumor, and cytoprotective activities, and analyzing the possible benefits to derive from their dietary ingestion, as well as, their pharmacological and therapeutic potential.

### 2.7. Fatty Acid Composition

The fatty acid compositions were determined where the results are shown in Table 3. The PGO was rich in polyunsaturated fatty acids (PUFA) (87.51%). The PUFAs were essential fatty acids, namely, linoleic acid 18:2 (omega 6), linolenic acid C18:3 (omega 3), as well as a bioactive conjugated punicic acid (CLnA, omega 5), and eicosapentaenoic (20:5 omega 3) and were 5.24, 0.18, 81.29 and 0.8, respectively. Punicic acid was determined by comparing with the standard of the GC retention time and mass spectra. It was 81.29% of total fatty acids in pomegranate seed oils. This was following the previous reports on punicic acid in pomegranate seed oils [13,19].

The PGO contained an extreme amount of CLnA than other well-known CLnA-rich seeds, which recommended the PGO to act a reduced impact against cancer, tumors, and obesity [20]. The total saturated fatty acids (SFA) of the present PGO recorded at 5.78% of total fatty acids. A total of 95% SFA in the PGO were palmitic acid (2.57%) and stearic acid (2.07%), which were higher than cultivars grown in Turkey and lower than Georgian cultivars [13,19]. Palmitoleic, oleic, vaccenic, gondoic, erucic, and nervonic acid were the monounsaturated fatty acid (MUFA) detected in the PGO and accounted for 6.71% of total fatty acids (Table 3). The MUFA content in the PGO was comparable to that in Georgian and Turkish cultivars [13,19]. However, the PGO contained much less oleic acid than red raspberry seeds (12.4%), blueberry (22.9%), onion (24.8–26%), parsley (80.9–81%), cardamom (49.2%), and pumpkin (36.3%) [21,22].

### 2.8. Encapsulation of the PGO

The diameter of the encapsulated PGO granules was determined using the electron microscope and Zeta potential values. The recorded particle size varied between (270 to 715 nm) for micro construction and between (14 to 16 nm) for nano construction. The medium size of the encapsulated particle for the PGO was teeny in nano-construction (between 14 to 16 nm), while the size variation was wide in micro-construction (between 270 to 715 nm). The zeta potentials values of micro- and nano-emulsion also were recorded at (−16.12 ± 0.34 mV) and (−24.41 ± 0.18 mV), respectively. Nano-capsules of the PGO appeared more disperse than the microcapsules that reflect a more stable stat. The high dispersion of the PGO nano-capsules may associate with their flat particle surface more than the micro-particle, which is known as less tendency for creaming with a chance of aggregation possibility.

### 2.9. Antimicrobial Activity of the PGO Types

The antimicrobial activity of the PGO against six bacterial strains showed an increment efficacy by the capsulation process. According to the data presented in Table 4, the G- bacteria was less sensitive to the PGO doses than their counterpart in the G+ bacteria did. The sensitivity of bacterial strain against the components of antioxidants joined to their influence on the cell walls [23,24,25]. The rise in the PGO performance was illustrated through the capability of the capsule wall to protect the bioactive components against loss or damage [26].

### 2.10. Mycotoxin Reducing of The PGO

The PGO impact of mycotoxin reduction was evaluated as limiting the capability of fungi to excrete their toxin. Mycotoxin in each media filtrate was determined and compared to the control after discarding the fungal growth. For the AFs; the inhibition ratio of the PGO ranged between 14 to 29.7%, while this ratio increased dramatically in case using the PGO capsules added to the growth media (Figure 2). It was ranged from 49.5 to 53% for PGO microcapsules and from 57 to 61% for PGO nanocapsules. Otherwise, the zearalenone toxin (ZEA) excretion reduced by 38.3, 62.7, and 78.5% for the PGO, microcapsules, and nanocapsules, respectively. It is worth mentioning that inhibition ratios of the fungal mycelial growth for some strains that counted as mycotoxin producers recorded at 17.4%, 12.1%, and 25.6% for *A. flavus* ITEM 698, *A. parasiticus* ITEM 11, and *F. culmorum* KF 846, respectively.

### 2.11. Cytotoxicity and Protection Effect of the PGO

The cytotoxic effect of the PGO was investigated in vitro on an isolated murine hepatocytes cell line using the MTT assay. The powerful EC100 value indicates the high safety of the PGO. This result shows that no significant difference observed between the EC100 of the PGO, and these doses were higher. The assay results indicated that PGO failed to show any cytotoxic effect on the isolated murine hepatocytes cell line at the recommended dosage level, and the EC100 value was recorded at 0.0288 ± 0.002 µg/mL.

Moreover, the protective effect of the PGO against aflatoxin-induced hepatotoxicity showed in Figure 3. The result illustrates that PGO possessed a higher significant protection effect (>63%) against aflatoxin-induced hepatotoxicity. The hepatotoxicity percentage in aflatoxin-exposed cells decreased by 63% when the AFB_1_ was mixed with the PGO oleoresin before injected into the cell line media. In Figure 3A, concentrations of the PGO, PGO+AFB_1_, and AFB_1_ lonely were applied in the cell line media. The AFB_1_ is known to occur carcinogenicity and cell death, which appeared in the figure. The results have expressed the amelioration that happened to the cell viability by adding PGO oleoresin to cell line media. However, the application of just PGO oleoresin reflects a few impacts on the cell line viability. This result has confirmed in Figure 3B, where the anti-hepatotoxic dose recorded as 0.028 and 0.184 µg/mL for the PGO and PGO–AFB_1_ applied in cell line media, respectively. These results reflect the bioactive role of the PGO generally and against mycotoxins in a specific.

Based on the fact that plant extracts (polar or non-polar) possessed bioactive molecules in various amounts, which previously were reported by a capability to reduce mycotoxin excretion [27,28]. The obtained results will be discussed in this section to prove the synergized impact of PGO oleoresin in minimizing the mycotoxin formation. The minimizing action encouraged using the PGO, which occurred by the insertion of PGO in the fungal growth of liquid media. The mycotoxin decreases compared to the control reflect how the PGO components were synergized. This change in the condition of the growth media did not encourage mycotoxin secretion. The AFs are groups of health-hazard compounds, and also are classified as polyketide-derived [29]. Their production relies on more than twenty genes clustered together in the DNA-sequence region. The genes encode a plurality of enzymes participatory in toxin synthesis as transcription agents [30]. A nearer strategy of synthesis process was reported for the ZEA toxins, where it is also having a lacto-coumarin ring connected to their toxicity. Subsequently, the synthesis of enzymes related to these genes could be affected by the existence of PGO active molecules.

The impact of plant phytochemicals on the aflatoxin gene cluster of toxigenic fungi and the biosynthetic pathway of aflatoxin was reported in several previous investigations. For instance, the expression of aflatoxin genes was reported downregulating in a presence of essential oils [31]. In addition, the phytochemicals, such as phenolic acids, are known to suppress gene expression in the aflatoxin pathway [32,33]. Moreover, the plant phytochemicals represent pivotal components that command the gene regulation in toxigenic fungi [34].

In specific, oxidizing agents can induce AF biosynthesis [35]. By the incidence of oxidative stress, the fungal molecules and enzymes are generated concomitantly to the AF biosynthetic gene cluster [36]. Therefore, it was also proposed that toxin production could also be acting to blend oxygen atoms and defend cells from oxidative damage through fungal life [37]. Otherwise, the plant phytochemicals of antioxidant molecules act as antifungals and affect the biosynthetic pathway of the aflatoxin gene cluster. It could serve out the limitation of aflatoxin biosynthesis by scavenging the factors that lead the fungi to feel oxidative stress, which may redirect the biological activity of fungi [38,39].

Indubitably, several plant types of enzymes had a capacity diminishing aflatoxin secretion due to the genetic impact on the AF cluster gene [40,41]. The effect of plant molecules on aflatoxin secretion varied according to the plant-active molecule. The AFB_1_ altered in diverse behaviors, such as double-bond eradication from a furan ring plus the readjustment for the lactone-ring structure to reproduce less toxic compounds [42].

Mycotoxins mainly destroy the cell system through their protein adducted or DNA-adducted action. The active form that reportedly attacks the cell-system is the 8–9 epoxide derivative [29,43]. This derivative is distinguished by the unsaturated bonds, which are classified as a principle step for cell mutations occurred. The fungal cells betake through several transactions during their life cycle. Pivotal stages are related to the fungal suffering of the stresses such as oxidative tension [6]. This stress could procure fungal mutations leading to changes in hormone and enzyme creation of the cell life-cycle [44,45]. In addition, oxidative stress is considered a significant factor of aflatoxin synthesis by fungi; it mainly deems the principle reason for aflatoxin biosynthesis by the excitation of the aflatoxin gene cluster. Oxidative stress of fungal cells is the playmaker that causes the cell conversion to produce aflatoxin [39,46]. As well, the AFs’ creation by fungal cells requires specific enzymes, which are approximately 18 enzyme types that are significant to complete the AF creation steps [38,47]. Most of these creation steps depend on the monooxygenation reactions. The presence of bioactive components possessed an antioxidant function leads for suppressing the aflatoxin creation [24,39,40]. This action was achieved by scavenging the free radicals, due to the PGO oleoresin bioactivity. The interaction between bioactive molecules presented by the PGO in growth-media and fungal metabolic reactions that lead to an inhibition of the mycotoxin production reflected as mycotoxin amount decreases. This action also may illustrate through the suppression of enzymes’ secretion of mycotoxin creation [41].

Several physiological proceedings in fungi cells are organized by oxidative changes, including the secretion of secondary metabolism components. In particular, the presence of oxidants and free radicals in the growth media are capable encourages the biosynthetic of AFs molecules [25]. In this regard, the abundant tocol derivatives (mainly gamma fractions) of oil can synergize antioxidant phenolics that also were existed in oil to change the condition media of fungal growth. These changes offer a favorable condition to fungi, which affected the fungal metabolism and converting to limit the mycotoxin formation.

In light of the model created by Kenne et al. [47], the aflatoxin biosynthesis by the fungal cell was to protect their cells against the reactive oxygen species accumulation, since the mono-oxygenase reactions are the prime response that regulates the conversion into aflatoxin in fungal cells [48,49]. The occurrence of oxidative stress and its related factors deems the kickoff point that forced the fungi for the mycotoxin formation. In this case, the mycotoxin creation reaction will help the fungal cell to scavenge its content of the free radicals and also to mitigate the oxidative stress influences [47,50]. This point could explain the cell transformation between primary and secondary metabolism production, where the latter occurs under the condition of oxidative stress [37,51,52]. The existence of the free-radicals in the fungal-media conducts the fungal strain to mitigate the unsuitable conditions and convert to excrete secondary metabolites [47,51].

It was pointed out that AF production could be a format to mingle atoms of oxygen to protect living-cell against oxidative damage [23]. In this regard, if there is a source of other molecules that are available in fungal medium and assist the scavenging free-radicals from it [37,39,53]. This action will be limited to the fungal oxidative stress, and the strain may be discontinued secretion of mycotoxins. The active molecules, which possess that action, was expressed by phytochemical antioxidants that are comprehensive the tocotromanols, phenolic acids, and carotenoids. These molecules are stimulated concomitantly and interact with the AF gene cluster throughout the AF biosynthesis process by the fungal cells [22,54]. This influence extended to the fungal cells for stopping secondary metabolites production or redirection to the vegetative growth [24,25,55].

Technically, by the PGO additive to the fungal growth media of filamentous fungi, the contents of PGO oleoresin interacts and fuses into the fungal media content and offers wide varieties of bioactive molecules [4,16]. The phenolic antioxidants, tocopherols, and tocotrienols represent the principle scavengers, which when present in fungal media, could mitigate the oxidative stress action on the fungal growth [3,24,39]. Moreover, it looks like PGO components represent a part of media micro-nutrients for the fungi, which affects the biochemical processes inside the fungal strain [13]. The mechanism, where mycotoxin has secreted, was influenced by the PGO presence and is related to the interaction between the PGO antioxidant potency and the free radical in fungal media. The antioxidant activity resulted from antioxidant phenolics and tocols components that account for the mitigation of fungal oxidative stress, which is finally reflected as a decrement in mycotoxin production of AFs or ZEA compounds.

## 3. Conclusions

The PGO oleoresin revealed a wealth of bioactive molecules such as phenolic compounds, sterols, and possessed an antioxidant potency. For instance, TPC and TFC were recorded at 39.06 ± 1.47 and 12.4 ± 0.96 mg/g, respectively. Moreover, tocochromanols (vitamin E) recorded richness, particularly for δ-tocopherol (414.6 ± 2.31 ug/g) and γ-tocotrienol (462.7 ± 3.05 ug/g). Carotenoids and sterols were designated by a substantial amount. The phenolic acids in the PGO are distinguished by their content of rutin (148.33 ± 0.25 mg/100 g), quercetin (135.83 ± 0.31 mg/g), and caffeic acid (115.37 ± 0.30 mg/g) compared to other vegetable-oil sources, while the principal phenolics are gallic, catechin, chlorogenic, and caffeic acids. The oleoresin showed to be rich in omega and conjugated fatty acids, where the prime fatty acid is punicic. This fatty acid is also known for potent bioactivity. Regarding the mycotoxin reduction ratio, PGO reduced the AFs and ZEA at up to 29.7% and 38.3%, respectively. The encapsulation technique utilized for the PGO, as micro and nano-construction, which achieved protection for the oleoresin bioactive components. This was reflected as an amelioration of mycotoxin reduction efficiency and may facilitate the PGO oleoresin application. This leads to more implementation of the oleoresin as a food supplement, with safety properties for the final product. While PGO oleoresin failed to show any cytotoxic effect on the isolated murine hepatocytes cell line, it has a protective effect against aflatoxin-induced hepatotoxicity that was recorded by (>63%). In this regard, the results pointed out the potency of PGO oleoresin in degradation influence on microbial contamination of toxigenic fungi and their metabolites. The suppression mechanism could be explained through a significant correlation between the enrichment of antioxidants in the media and the fungal metabolism redirection. This also may lead to recommending PGO oleoresin as an additive in food preservation.

## 4. Materials and Methods

### 4.1. Source and Extraction of Pomegranate Seed Oil

The Manfalouty seeds variety was obtained as a coproduct during pomegranate-juice extraction. The PGO performed (35 °C/45 min) using petroleum ether (40:60) by an ultrasonic assessed method (Ultrasonic UP650 probe, Acculab Inc., East Brunswick, NJ, USA). The collected emulsion was evaporated by rotary evaporator (40 °C), then oil was filtered, packed in amber bottles, and stored (−20 °C).

### 4.2. Determination of Total Phenolic Compounds (TPC)

The phenolic content of oil was extracted the same as Ramadan et al. [56] with modifications. In a falcon tube, one gram oil dissolved using 3 mL n-hexane, then 5 mL aqueous methanol (80%) was added before shaking (3 min/vortex). The tube was rested for a minute, then centrifugation (4200 g/10 min); the hydro-alcoholic layer separated using a Pasteur pipette. The extraction process was repeated twice, and collected extracts were combined. The polar extract was dissolved in 10 mL acetonitrile and was washed using n-hexane (10 mL/3 times). The purified acetonitrile-phenolic extract was vacuum-evaporated to near dryness by the rotary (35 °C), finally dissolved by aqueous methanol. About 0.2 mL of extracted solution and diluted Folin–Ciocalteu’s phenol reagent (1 mL) were mixed. After 3 min, sodium carbonate (0.8 mL/7.5%) was added, the mixture was rested for 30 min, and absorbance was measured using a UV spectrophotometer (Shimadzu, Kyoto, Japan) at wavelength 760 nm. The TPC measured and expressed as mg gallic acid equivalent (GAE)/100 g of the fresh sample against the blank.

The total flavonoid content (TFC) of the PGO was determined using an aluminum chloride colorimetric method [57]. Briefly, 1 mL aqueous methanol extract (1 mg/mL; 80% methanol) or standard catechol solution was mixed with one milliliter of the AlCl3 (2% *w*/*v*) in methanol. The absorbance against blank was measured at 440 nm using UV spectrophotometer (after incubating for 40 min/25 °C). The TFC calculated using the catechol calibration curve; results expressed as catechol equivalent (CE) per gram of the PGO, where the samples performed in triplicate.

### 4.3. Determination of Tocopherols and Tocotrienols Content

Tocopherols and tocotrienols were evaluated due to the method described by Balz et al. [58]. Diluted oil injected into high-performance liquid chromatography (HPLC), sample-volume-adjusted at 20 µL, the mobile phase was n-hexane: tertabutyl-methyl ether (96:4 *v*/*v*), the isocratic system applied with 1.0 mL/min flow and wavelength of 295 nm, using a UV detector.

### 4.4. Determination of Sterols and Carotenoid Content

The oil content of sterols was measured due to the methodology described by Stuper-Szablewska et al. [59]. While, carotenoids separated and their quantity in samples was evaluated using Acquity ultra-high performance liquid chromatography (Waters, Milford, MA, USA) according to a method and conditions described by Kurasiak-Popowska et al. [60].

### 4.5. Antioxidant Activity Assays

#### 4.5.1. Scavenging Activity of DPPH Radicals

Antioxidant activity was determined using the 2, 2-diphenyl-1-picrylhydrazyl (DPPH) free radical scavenging assay according to the method of Ishtiaque et al. [61] with some modifications. Briefly, different concentrations of oil (dissolved by 3 mL toluene) were mixed with 1 mL of 0.078 mM DPPH dissolved in toluene as a toluene solution. The mixture was shaken well and allowed to stand (25 °C/30 min/in the dark). The absorbance was measured at 517 nm using a UV spectrophotometer. The tests were carried out in triplicates. The DPPH radical inhibition calculated by the following equation:$$\%\mathrm{Inhibition}= [\frac{\mathrm{Abs}\;517\;\left(\mathrm{control}\right)-\mathrm{Abs}\;517\left(sample\right)}{Abs\;517\left(control\right)}]\times 100$$

#### 4.5.2. Determination of Reducing Power

The reducing power of oil was determined according to the method of Yen and Duh [62] with modifications. One milliliter of toluene containing (7.81–1000 µg of oil) was mixed by phosphate buffer (2.5 mL, 0.2 M, pH 6.6) and potassium ferric-cyanide (2.5 mL, 1%); the mixture was incubated (50 °C/20 min). At an incubation time ended, 2.5 mL trichloro-acetic acid (10%) was added to the mixture and centrifuged (3000 g/10 min). The upper layer of solution (2.5 mL) was mixed with distilled water (2.5 mL) then FeCl3 solution (0.5 mL/0.1%) was added. The absorbance was measured at 700 nm, and reducing power was expressed as ASE/mg. The ASE means equivalent power of 1 mg sample (E) that reducing the power of 1 nmol ascorbic acid (AS).

### 4.6. Determination of Phenolic Fractions

The phenolic fraction compounds in the PGO aqueous methanolic extract were determined according to the methodology described by Stuper-Szablewska et al. [59]. The analysis was performed utilizing an Acquity H class UPLC system equipped with the Waters Acquity PDA detector (Waters, Milford, MA, USA). Chromatographic separation was accomplished using an Acquity UPLC^®^ BEH C18 column (100 × 2.1 mm, 1.7 µm) (Waters, Dublin, Ireland). The elution was achieved using the following mobile phases in the appropriate gradient: A: acetonitrile with 0.1% formic acid, B: 1% aqueous formic acid (pH = 2.0). The phenolic compound concentrations were determined at λ = 320 and 280 nm and were identified based on a retention time comparison of the analyte peak by a retention time for standards and by adding a specific amount of the standard to the analyzed samples and repeating this analysis. The limit of detection was one microgram per gram of the sample.

### 4.7. Determination of Fatty Acid Profile

The PGO fatty acid content was evaluated using the methodology described by Abdel-Razek et al. [27]. Briefly, diluted oil was analyzed by Agilent 7890 apparatus (Agilent Technologies, Santa Clara, CA, USA) supported by the FID and capillary Innowax column (30 m × 0.20 mm × 0.20 mm). A flow rate of carrier gas was 1.5 mL/min, and the column temperature was 210 °C. The results were recorded as weight percentages after integration and calculation using the Chem-Station and comparing the retention times with authentic standards. 

### 4.8. The PGO Encapsulation

Maltodextrin (MD) dissolved at a rate of 10% in distilled water (50 °C/2 h) to equip the primary solution, while soy protein (SP) was dissolved wholly in water (8%; 60 °C/30 min) as the secondary solution. The mixed solution of encapsulation was prepared as 2 MD: 1 SP (*v*/*v*). Afterward, adding a quantity of 1% Arabic gum solution, then stirred for 30 min until the whole homogenization. The PGO was loaded gently with oil-droplets dripped into the solution mixture during stirring for 10 min using an Ultra-Turrax probe (T10 basic ULTRA-TURRAX^®^). The nanoemulsion was prepared by a treatment of the resultant solution using an Ultrasonic probe for 45 min/30 °C. The transmitted electron microscope capture of the PGO micro and nanoemulsion were displayed in Appendix A (Figure A1).

### 4.9. Determination of Antimicrobial and Antifungal Activity

The antimicrobial activity of the PGO, micro-, and nano-capsule emulsion were determined by measuring the minimal inhibiting concentration according to the methodology of Abdel Razek et al. [27]. Six bacterial strains (*Staphylococcus aureus* ATCC33591, *Bacillus cereus* ATCC 11778, *Listeria monocytogenes* ATCC 19111, *Escherichia coli* ATCC 11229, *Shigella sonni* ATCC 29930, and *Salmonella typhi* ATCC 14028) were reactivated from lyophilized stocks on tryptic soy agar. Three fungal strains were utilized to evaluate the mycotoxin reduction. These strains were *A. flavus* ITEM 698, *A. parasiticus* ITEM 11, and *F. culmorum* KF846. The strains were brought from an agro-food microbial culture collection, ITEM microbial culture collections, ISPA, CNR, Italy; and the *F. culmorum* KF846 was obtained from the plant pathology laboratory, Institute of Plant Genetics, Polish Academy of Science, Poznan, Poland.

### 4.10. Estimation of Antibacterial and Antifungal Activity

Oil emulsion prepared and diluted with phosphate buffer saline (PBS) at pH equal to 7.2 ± 0.1. The same number of subsequent concentrations performed. The minimal concentration of the emulsion that inhibits microbial growth was determined using the microdilution method by serially diluted in sterile nutrient broth [63].

### 4.11. The PGO Impact on Mycotoxin Excretion in Liquid Media

In conical flasks of 500 mL, a volume of Czapek-Dox media equal to 150 mL was autoclaved for the evaluation of the PGO impact on fungal production of mycotoxin. Two strains (*A. flavus* ITEM 698 and *F. culmorum* KF846) were cultured individually in the flasks in both presence and absence of the PGO in media. The strain of *A. flavus* is known to secrete four types of the AFs, while *F. culmorum* strain is known to secrete zearalenone (ZEA). Likewise, toxins concentration in a filtrated media after incubation (28 °C/5 days) were evaluated using the HPLC apparatus according to the methodology described by Shehata et al. [64].

### 4.12. Mycotoxin Determination

High-performance liquid chromatography, Agilent 1100 (Agilent Technologies, Hewlett-Packard Strasse 876,337 Waldbronn, Germany), was used for AFs determination. The mobile phase was water: acetonitrile: methanol (6:3:1). The chromatographic separation was performed with an Extend-C18, Zorbax column (250 mmx 5 µm, Agilent Co.). The column temperature was 40 °C, and the flow rate was 1.0 mL/min; the injection volume was 20 µL for samples and standard. The detector adjusted at 360/440 nm for the excitation and the emission wavelength, respectively. Data were integrated and recorded using a Chem-Station software Manager Hewlett-Packard.

For zearalenone (ZEA), it was extracted and evaluated from media similar to the methodology of Badr et al. [65]. The mobile phase was a mixture of acetonitrile: water: methanol (46:46:8, *v*/*v*), the flow-rate was 1.0 mL/min. Quantification of the ZEA was performed by comparing the retention time against the standard. The identity of the ZEA was confirmed at 274 and 440 nm for excitation and emission wavelengths, respectively, compared to the ZEA standard peak.

### 4.13. Detecting the PGO Prevention against The AFB_1_ Impact

Normal hepatocytes were isolated from Albino mice, according to the method of Whitehead and Robinson [66], with modifications. After mincing liver tissue, it was incubated in a collagenase I solution (3 U/mL), gentle shaking (30 min), then centrifuged (1000 g/5 min). Cell pellets suspended in William’s E medium (supplemented with 10% PBS) and incubated at 37 °C using a CO_2_ incubator (5%). After 90% cell confluent, hepatocytes were passaged with trypsin-EDTA.

Hepatocytes were seeded as 1 × 10^4^ cells per well, in a 96-well cell culture plate, and re-incubated. After cell attachment (24 h), serial concentrations of tested materials (PGO/AFB_1_/PGO-AFB_1_) were injected into tested wells and the cell hepatocytes were incubated (72 h). The cell viability was assayed by the MTT method [67]. Twenty microliters of the MTT solution (5 mg/mL; Sigma, St. Louis, MI, USA) was added to each well and incubated (37 °C/3 h). The MTT solution was removed then 100 µL DMSO was added, the absorbance was measured with a microplate reader (BMG LabTech, Offenburg, Germany) at 570 nm wavelength. The GraphPad Instat software estimated the safe doses (EC100) of the PGO, and the growth inhibition (cytotoxicity) percentage was calculated using the following equation:
$$\%\;\mathrm{Cytotoxicity}\;=\; [100\;-\;\left(\mathrm{Ae}/\mathrm{Au}\right)]\;\times \;100.$$
where:Ae: absorbance of exposed cells.Au: absorbance of unexposed control cells.

### 4.14. Statistical Analysis

The results were expressed as mean values ± standard deviation from at least three replicates. The statistical analyses of data were performed using GraphPad Prism 7 (GraphPad Software Inc., San Diego, CA, USA).

## Figures and Tables

**Figure 1 toxins-12-00748-f001:**
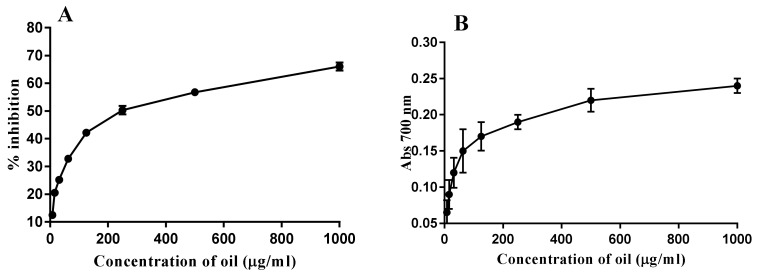
Antioxidant activity of pomegranate oil (**A**) DPPH—2, 2-diphenyl-1-picrylhydrazyl; (**B**) (reducing power).

**Figure 2 toxins-12-00748-f002:**
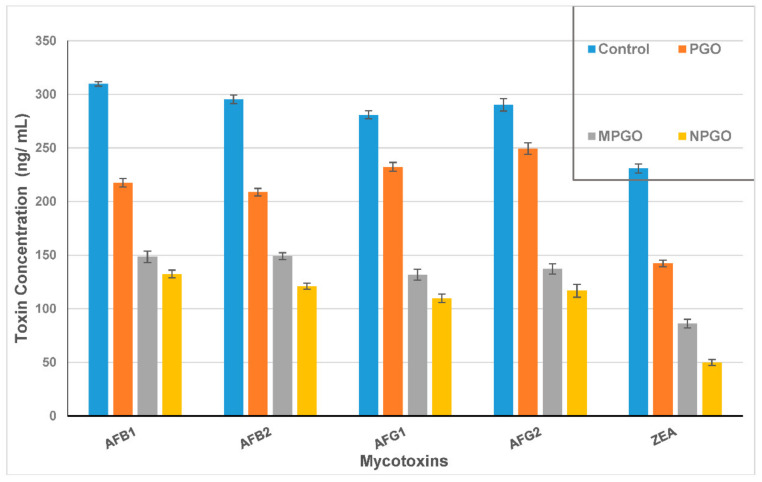
Aflatoxin and zearalenone reducing in Czapek Dox media in a presence of the pomegranate oil Emulsions. AFB_1_: aflatoxin B_1_; AFB_2_: aflatoxin B_2_; AFG_1_: aflatoxin G_1_; AFG_2_: aflatoxin G_2_; and ZEA: zearalenone toxin. PGO: pomegranate oil; MPGO: micro-emulsion of pomegranate oil; NPGO: nanoemulsion of pomegranate oil.

**Figure 3 toxins-12-00748-f003:**
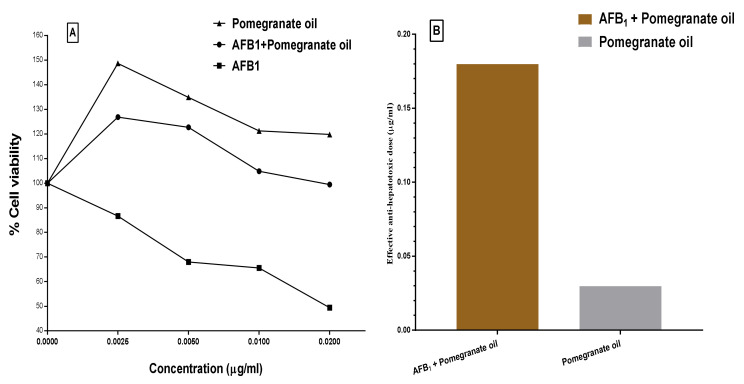
The anti-hepatoxicity effect of pomegranate oil and pomegranate oil + AFB_1._ (**A**) Viability of pomegranate oil and pomegranate oil + AFB_1_, (**B**) effective anti-hepatotoxic doses of pomegranate oil and pomegranate oil + AFB_1_.

**Table 1 toxins-12-00748-t001:** Biochemical composition of pomegranate oil.

Biochemical Parameter	Pomegranate Oil
Total phenolic (mg GAE/g)	39.06 ± 1.47
Total flavonoid (mg CE/g)	12.4 ± 0.96
Tocochromanols (μg/g)	
α-tocopherol	16 ± 2
β-tocopherol	0.4 ± 0.42
δ-tocopherol	414.6 ± 2.31
γ-tocopherol	13.7 ± 1.53
α-tocotrienol	1.6 ± 0.15
β-tocotrienol	0.67 ± 0.12
δ-tocotrienol	2.03 ± 0.21
γ-tocotrienol	462.7 ± 3.05
Total	911.7 ± 9.79
Carotenoids (µg/g)	
Lutein	188.3 ± 1.79
Zeaxanthin	16.47 ± 0.21
β-Carotene	0.54 ± 0.078
Total	205.31 ± 2.08
Sterols (mg/100g)	
Campesterol	55.95 ± 2.55
Stigmasterol	0.11 ± 0.029
β-sitosterol	1573.47 ± 3.47
Brassicasterol	0.023 ± 0.009
Campestanol	0.077 ± 0.021
Delta-5 Avenasterol	9.93 ± 0.125
Delta-7 Stigmastenol	0.273 ± 0.012
Total	1639.36 ± 6.22

Values represent the mean ± standard deviation of three independent replicates (*n* = 3); GAE: gallic acid equivalents; CE: catechin equivalents.

**Table 2 toxins-12-00748-t002:** Phenolic compounds of the pomegranate oil.

Phenolic Compounds mg/100g	Average
Apigenin	26.53 ± 0.35
Catechin	396.63 ± 0.45
Kaempferol	85.5 ± 0.3
Luteolin	73.97 ± 0.15
Naringenin	27.07 ± 0.35
Quercetin	135.83 ± 0.31
Rutin	148.33 ± 0.25
Vitexin	31.3 ± 0.26
4-hydroxybenzoic	72.4 ± 0.17
Caffeic	115.37 ± 0.30
Chlorogenic	175.57 ± 0.31
Ferulic	0.013 ± 0.005
Gallic	784.13 ± 0.25
p-Cumaric	12.67 ± 0.23
Protocatechuic	99.33 ± 0.15
Sinapic	ND
Syringic	0.017 ± 0.011
t-Cinnamic	13.4 ± 0.3
Vanillic	6.37 ± 0.21
Vanillin	42.43 ± 0.31

Values represent the mean ± standard deviation of three independent replicates (*n* = 3); ND: not detected.

**Table 3 toxins-12-00748-t003:** Fatty acid composition of the pomegranate oil.

Synonym	Fatty Acid	%
C12:0	Lauric acid	ND
C14:0	Meristic Acid	0.02
C16:0	Palmitic Acid	2.57
C16:1	Palmitoleic Acid	0.1
C17:0	Margaric Acid	0.14
C18:0	Stearic Acid	2.07
C18:1	Oleic Acid	4.33
C18:1 trans	Vaccenic Acid	0.64
C18:2	Linoleic Acid	5.24
C18:3	Linolenic Acid	0.18
C18:3 n5	Punicic Acid	81.29
C20:0	Arachidic Acid	0.46
C20:1	Gondoic Acid	0.56
C20:1	Gadoleic Acid	ND
C20:5	Eicosapentaenoic acid	0.8
C22:0	Behenic Acid	0.13
C22:1	Erucic Acid	1.05
C24:0	Lignoceric acid	0.39
C24:1	Nervonic Acid	0.03
∑ SFA	Sum of saturated fatty acids	5.78
∑ MUFA	Sum of monounsaturated fatty acids	6.71
∑ PUFA	Sum of polyunsaturated fatty acids	87.51
MUFA/PUFA	Fractionation	0.077
SFA/PUFA	Fractionation	0.061

**Table 4 toxins-12-00748-t004:** The minimal bacterial concentration of the PGO types.

	PGO (mg/mL)	MPG (mg/mL)	NPGO (mg/mL)	Azithromycin (mg/mL)	Note
*Escherichia coli* ATCC 11229	220	120	70	0.015	(G-) bacteria
*Salmonella typhi* ATCC 14028	130	80	50	0.016	(G-) bacteria
*Shigella sonni* ATCC 29930	160	90	60	0.016	(G-) bacteria
*Staphylococcus aureus* ATCC 33591	180	100	50	0.012	(G+) bacteria
*Listeria monocytogenes* ATCC19111	150	90	50	0.012	(G+) bacteria
*Bacillus cereus* ATCC11778	150	90	50	0.012	(G+) bacteria

PGO: pomegranate oil; MPGO: microcapsuled pomegranate oil; NPGO: nanocapsuled pomegranate oil; azithromycin was applied as a standard antibiotic reference.

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
