# Peer review of "Bioactive Components of Pomegranate Oil and Their Influence on Mycotoxin Secretion"

_toxins, 2020, doi:10.3390/toxins12120748_

Round 1

Reviewer 1 Report

The authors presented a very interesting topic and presented the results very well. But, I doubt that you introduced and discussed the results very well. Please revise according to the comments in the PDF version attached.

Author Response

  • For Reviewer 1 comments:
  • Extensive editing of English and style.

  Response: The authors review the manuscript and change, correct the mistakes that displayed.

  • The introduction must be improved? 

   Response: The authors reviewed the introduction and make the changes as requested in the attached PDF file.

  • The conclusion needs some improvements.

 Response: the conclusion was rewritten again to be as good as requested.

  • A discussion needs more enhancement. 

  Response: the changes were done to enhance the discussion as it was declared in the PDF file attached.

Reviewer 2 Report

I found this submission as a very interesting one.
The language has minor typing mistakes, but I am not an English native speaking person to judge it.
Overall, I have a major comment, and a few minor ones:

Major: authors mention antymycotoxin secreting activity, but have not measured fungal growth. So, I cannot judge if the oil acted by inhibiting fungal growth, or acted by inhibiting mycotoxin secretion. By measuring the ratio mycotoxin / fungal biomass, one may conclude if there is inhibition, or even stimulation of mycotoxins.

Authors should consider this comment in a possible revised version.

Minor: authors used data with a lot of significant digits. For instance, in line 109 they reported 66.03 %; how are you so sure it is 66.03 %, and not just 66 %?
Use an appropriate abbreviation for zearalenone. ZN is not a common one.

Author Response

Response for the comments of the reviewer 2

  • Minor English revision. 

  the Response: Authors have reviewed the manuscript and correct the discovered mistakes.

  • Major: authors mention antimycotoxin secreting activity, but have not measured fungal growth. 

    Authors should consider this comment in a possible revised version.

 The response: this part of the study was corrected and was added as 

It is worth mentioning that; inhibition ratios of the fungal mycelial-growth for some strains that counted as mycotoxin-producers were recorded at 17.4%, 12.1%, and 25.6% for A.flavus ITEM 698, A. parasiticus ITEM 11, and F. culmorum KF 846, respectively."

  • Minor: authors used data with a lot of significant digits. For instance, in line 109 they reported 66.03 %; how are you so sure it is 66.03%, and not just 66 %?

 The response: these results recorded as a calculation ratio, and in the correction step it was approximated as it was requested.

  •  Use an appropriate abbreviation for zearalenone. ZN is not a common one.

The response: this abbreviation was corrected to be ZEA instead of ZN.

Round 2

Reviewer 1 Report

All comments and suggestions have been addressed.

Reviewer 2 Report

Authors have answered my comments.